# A Physiologically Based Pharmacokinetic Model of Ketoconazole and Its Metabolites as Drug–Drug Interaction Perpetrators

**DOI:** 10.3390/pharmaceutics15020679

**Published:** 2023-02-17

**Authors:** Fatima Zahra Marok, Jan-Georg Wojtyniak, Laura Maria Fuhr, Dominik Selzer, Matthias Schwab, Johanna Weiss, Walter Emil Haefeli, Thorsten Lehr

**Affiliations:** 1Clinical Pharmacy, Saarland University, 66123 Saarbruecken, Germany; 2Dr. Margarete Fischer-Bosch-Institut of Clinical Pharmacology, 70376 Stuttgart, Germany; 3Departments of Clinical Pharmacology, and of Biochemistry and Pharmacy, University Hospital Tuebingen, 72076 Tuebingen, Germany; 4Cluster of Excellence iFIT (EXC2180) “Image-Guided and Functionally Instructed Tumor Therapies”, University Tuebingen, 72076 Tuebingen, Germany; 5Department of Clinical Pharmacology and Pharmacoepidemiology, University of Heidelberg, 72076 Tuebingen, Germany; 6German Center for Infection Research (DZIF), Heidelberg Partner Site, 69120 Heidelberg, Germany

**Keywords:** physiologically based pharmacokinetic (PBPK) modeling, ketoconazole, cytochrome P450 3A4 (CYP3A4), P-glycoprotein (P-gp), reversible inhibition, metabolites, drug–food interaction, drug–drug interaction

## Abstract

The antifungal ketoconazole, which is mainly used for dermal infections and treatment of Cushing’s syndrome, is prone to drug–food interactions (DFIs) and is well known for its strong drug–drug interaction (DDI) potential. Some of ketoconazole’s potent inhibitory activity can be attributed to its metabolites that predominantly accumulate in the liver. This work aimed to develop a whole-body physiologically based pharmacokinetic (PBPK) model of ketoconazole and its metabolites for fasted and fed states and to investigate the impact of ketoconazole’s metabolites on its DDI potential. The parent–metabolites model was developed with PK-Sim^®^ and MoBi^®^ using 53 plasma concentration-time profiles. With 7 out of 7 (7/7) DFI AUC_last_ and DFI C_max_ ratios within two-fold of observed ratios, the developed model demonstrated good predictive performance under fasted and fed conditions. DDI scenarios that included either the parent alone or with its metabolites were simulated and evaluated for the victim drugs alfentanil, alprazolam, midazolam, triazolam, and digoxin. DDI scenarios that included all metabolites as reversible inhibitors of CYP3A4 and P-gp performed best: 26/27 of DDI AUC_last_ and 21/21 DDI C_max_ ratios were within two-fold of observed ratios, while DDI models that simulated only ketoconazole as the perpetrator underperformed: 12/27 DDI AUC_last_ and 18/21 DDI C_max_ ratios were within the success limits.

## 1. Introduction

The imidazole derivative ketoconazole is used topically for the treatment of dermal fungal infections and systemically as therapy for Cushing’s syndrome [1,2]. For systemic applications, ketoconazole is administered as oral tablets in dose ranges of 200–400 mg [3]. For doses below 400 mg, pronounced drug–food interactions (DFIs) have also been observed for the biopharmaceutical classification system (BCS) class II compound as its oral bioavailability is highly limited by its poor solubility of 0.006 mg/mL (at a pH of 7.5) [4,5]. In contrast, at higher doses, DFIs do not significantly modulate ketoconazole exposure [6].

Upon absorption, ketoconazole is mainly bound to albumin and blood cells, and only 1% is unbound in plasma [1]. Moreover, ketoconazole has been discussed as a substrate of the efflux transporter P-glycoprotein (P-gp) as well as a substrate of cytochrome P450 3A4 (CYP3A4), arylacetamide deacetylase (AADAC), and uridine diphosphate glucuronosyltransferase 1A4 (UGT1A4) [7,8,9,10,11]. Roughly 10–37% of unchanged ketoconazole is eliminated in feces, while 2–4% can be found in urine [5].

A systemic administration of ketoconazole for the treatment of fungal infections is not recommended, as oral ketoconazole intake might result in liver injury and can lead (similar to other azole antifungals) to prolonged QT intervals; therefore, an increased risk for torsades de pointes tachycardia [11,12]. The efficacy of ketoconazole in reducing cortisol levels for the treatment of Cushing’s disease may outweigh the risk of potential side effects [2]. The likelihood of experiencing severe adverse drug reactions can be further amplified due to ketoconazole’s strong drug–drug interaction (DDI) potential. Here, the exposure of the co-administered drug might be increased via inhibition of its drug metabolism [12,13].

Based on its strong DDI potential, ketoconazole is systematically administered in clinical studies as a DDI perpetrator drug [14,15]. Here, it serves as a potent inhibitor of CYP3A4 and P-gp, among other proteins, with substantial increases in drug exposure of victim compounds being reported in the literature. For example, the administration of 400 mg of ketoconazole over four days led to a 15-fold increase in the area under the plasma concentration-time curve (AUC) of midazolam [16], while pretreatment with 200 mg of ketoconazole twice daily over three days increased the AUC of triazolam 11-fold [17]. However, since ketoconazole inhibits CYP3A4 and P-gp reversibly and has a mean half-life of only 160 min (after a 400 mg dose) [18], long-term inhibitory effects cannot be explained by the involvement of the parent compound alone [19]. In the case of itraconazole, which is also an azole antimycotic as well as a potent DDI perpetrator drug, its metabolites contribute to its DDI activity; e.g., by reversibly inhibiting CYP3A4 [20]. Equivalently, some of ketoconazole’s inhibitory potential can also be attributed to its metabolites. One important metabolite, *N*-deacetylketoconazole (M1), which is formed via AADAC, was reported to inhibit the same enzymes and transporters as ketoconazole itself, including CYP3A4 and P-gp [13]. As for itraconazole, because three of its metabolites are involved in DDIs, it might be reasonable to assume that M1 is not the only metabolite responsible for ketoconazole-mediated DDIs; for example, the structurally similar *N*-deacetyl-*N*-hydroxyketoconazole (M2) among other metabolites might also contribute [7,21]. However, the individual contributions of ketoconazole’s metabolites to the observed DDI effects are still unknown, especially regarding their long-term inhibitory effects. To investigate the involvement of a perpetrator’s metabolites in DDIs, model simulations can be performed to assess their contributions and impact on their parent’s overall DDI potential. Here, physiologically based pharmacokinetic (PBPK) modeling can assist in testing hypotheses regarding the potential impacts of metabolites on ketoconazole’s strong observed DDI potential. The usefulness of parent–metabolite PBPK modeling for the investigation of drug interactions by imidazole derivatives was previously demonstrated in the application of an itraconazole–metabolites PBPK model within an extensive CYP3A4–P-gp–DDI network by Hanke et al. [22]. In general, the application of PBPK modeling is recommended by both the European Medicines Agency (EMA) and the U.S. Food and Drug Administration (FDA) for the different stages of the drug development pipeline [23,24,25].

Thus, the aims of the present study were (i) to build a PBPK model for ketoconazole under fasted and fed conditions and (ii) to investigate the contributions of its metabolites; i.e., M1 and M2, to ketoconazole’s DDI potential by predicting DDIs with the parent alone in comparison to DDIs with the parent alongside its metabolites as perpetrators impacting the pharmacokinetics of the victim drugs alfentanil, alprazolam, midazolam, triazolam (CYP3A4 victim drugs), and digoxin (P-gp victim drug). The developed parent–metabolites PBPK model files will be made publicly available at http://models.clinicalpharmacy.me.

## 2. Materials and Methods

### 2.1. Software

The PBPK model was developed with the open-source modeling software PK-Sim^®^ and MoBi^®^ (Open Systems Pharmacology Suite 11 released under the GPLv2 license by the Open Systems Pharmacology community; www.open-systems-pharmacology.org (accessed on 20 April 2022)) [26]. Published clinical study data were digitized using GetData Graph Digitizer 2.26.0.20 (© S. Federov) according to best practices [27]. Model input parameter estimation using Monte Carlo or Levenberg–Marquardt optimizations, by minimizing the sum of squares between the simulation and measurements from all included studies, and local sensitivity analyses were performed within PK-Sim^®^. Pharmacokinetic parameter analysis, model performance measures, and plots were compiled in R 4.1.3 (The R Foundation for Statistical Computing, Vienna, Austria) using RStudio 1.2.1335 (RStudio PBC, Boston, MA, USA).

### 2.2. Clinical Data

Clinical trials of ketoconazole with single-dose and multiple-dose regimens in fasted and fed participants were gathered and digitized from the literature [27]. Moreover, additional mean and individual plasma concentration-time profiles for ketoconazole and M1 were kindly provided by Weiss et al. [13]. Collected plasma concentration-time profiles were divided into a training dataset for model building and a test dataset for model evaluation. Studies in the training dataset were selected to include ketoconazole and M1 plasma concentration-time profiles and a wide ketoconazole dosing range administered in different formulations. No plasma concentration-time profiles of the metabolite M2 could be found in the literature. The compiled training and test datasets are documented in clinical study tables in Appendix A.

### 2.3. PBPK Model Building

The model-building process began with an extensive literature search for physicochemical properties and information about the absorption, distribution, metabolism, and excretion processes of ketoconazole and its metabolites.

Mean and mode demographic information (age, sex, ethnicity, body weight, and height) listed in clinical study reports was used to create virtual individuals for each study. If entries were partially missing; i.e., lacking information on weight or height, data were informed based on the suggested value provided by PK-Sim^®^ computed from the respective implemented population databases. If no data were available, a virtual standard individual with default values was created (see Appendix A).

Tissue distributions of enzymes were implemented according to the PK-Sim^®^ expression database and are listed in Appendix A.

Model parameters that either could not be adequately informed by the literature or were involved in important QSAR model estimates of permeability and distribution processes were optimized by fitting the model simultaneously to all plasma concentration-time profiles of the training dataset.

### 2.4. Drug–Food Interaction Modeling

The compiled clinical studies included data on ketoconazole administration under fasted or fed conditions. Data on particle size distributions were gathered from the literature to inform the parametrization of formulation models for oral ketoconazole solutions and tablets for simulations with and without the intake of food. To simulate the effect of DFIs on oral ketoconazole absorption, intestinal permeabilities were estimated based on the fasted and fed datasets, and gastric emptying time was adapted for the fed state. If no information about fasted or fed study conditions was provided, the fed state was assumed if (i) a delay in the time of maximum plasma concentration (T_max_) of more than two hours could be observed, (ii) multiple doses were administered within a day (as a continuous fasted state was considered unlikely), or if (iii) single doses as oral tablets of 800 mg or higher were administered (as differences in ketoconazole plasma exposure were found to be negligible for higher doses between fasted and fed states) [6].

### 2.5. Drug–Drug Interaction Modeling

To model the effect of DDIs, reversible inhibition of CYP3A4 and P-gp using ketoconazole and its metabolites was implemented into the parent–metabolites PBPK model using the respective in vitro data derived from the literature (if available). Previously published PBPK models of the CYP3A4 victim drugs alfentanil, alprazolam, midazolam, and triazolam as well as the P-gp victim drug digoxin were used to simulate DDI scenarios with ketoconazole co-administration [22,28,29]. The victim drug PBPK models were used to evaluate the performance of the ketoconazole model in DDI scenarios. Interaction partner models were selected if (i) the FDA listed them as sensitive or moderately sensitive substrates for CYP3A4 and P-gp [30], (ii) the evaluation as CYP3A4 and P-gp victim models was thoroughly investigated in DDI networks [22,31], (iii) models were developed in the Open System Pharmacology Suite, and (iv) model files were publicly available and accessible. Here, simulations were performed with and without the inclusion of ketoconazole metabolites.

### 2.6. PBPK Model Evaluation

Model evaluations included graphical comparisons of (i) predicted and observed plasma concentration-time profiles by plotting model predictions alongside their respective observed data, (ii) predicted and observed plasma concentration values in goodness-of-fit plots, and (iii) predicted and observed area under the plasma concentration-time curve calculated from the time of drug administration to the time of the last concentration measurement (AUC_last_) and maximum plasma concentration (C_max_) values. Additionally, as quantitative measures of the model performance, the mean relative deviation (MRD) of all predicted plasma concentrations and geometric mean fold error (GMFE) of all predicted AUC_last_ and C_max_ values were calculated according to Equations (1) and (2). Predictions with MRD and GMFE values ≤ 2 were considered successful model predictions.
(1)$${\mathrm{MRD}=10}^{x};\;x=\sqrt{\frac{\sum_{i=1}^{k}{\;(\log}_{10}{ĉ}_{i}{\;-\;\log}_{10}c_{i}{)}^{2}}{k}}$$
where c_i_ = the ith observed plasma concentration, ĉ_i_ = the corresponding predicted plasma concentration, and k = the number of observed values.
(2)$${\mathrm{GMFE}=10}^{x};\;x=\frac{\sum_{i=1}^{m}\left|\log_{10}\;\left(\frac{{\hat{\mathrm{PK}}}_{i}}{{\mathrm{PK}}_{i}}\right)\right|}{m}$$
where ${\hat{\mathrm{PK}}}_{i}\;$= the ith predicted AUC_last_ or C_max_ value, ${\mathrm{PK}}_{i}$ = the corresponding observed AUC_last_ or C_max_ value, and m = the number of studies.

Local model sensitivity to single parameter changes was analyzed for the AUC of ketoconazole and M1 after multiple dose administrations in fasted and fed states. Analyses included parameters that were either optimized or assumed to impact the AUC. Appendix A provides a detailed description of the performed local sensitivity analyses.

### 2.7. Drug–Food and Drug–Drug Interaction Model Evaluation

To assess the model performance of DFI and DDI effects, model predictions were evaluated with graphical comparisons of plasma concentration-time profiles, AUC_last_, and C_max_ values. Furthermore, effect ratios were calculated for the PK parameters AUC_last_ and C_max_ according to Equation (3):(3)$$\mathrm{DFI}\;\mathrm{or}\;\mathrm{DDI}\;\mathrm{PK}=\;\frac{{\mathrm{PK}}_{\mathrm{effect}}\;}{{\;\mathrm{PK}}_{\mathrm{reference}}}$$
where PK = PK parameter (AUC_last_ or C_max_) either of the DFI or DDI effect profile (${\mathrm{PK}}_{\mathrm{effect}}$) or of the respective control or placebo profile as the reference (${\mathrm{PK}}_{\mathrm{reference}}$).

In the case of DFIs, comparisons between fed (effect) and fasted (reference) conditions were only conducted for self-controlled studies with equal-dose regimens. In the case of DDIs, victim-drug plasma concentration-time profiles and PK parameters during co-administration of ketoconazole as the perpetrator drug (effect) were compared to respective measures without ketoconazole administration (reference).

As a quantitative measure of the effect model performance, *GMFE* values of the predicted AUC_last_ and C_max_ values as well as their effect ratios were calculated according to Equation (2).

## 3. Results

### 3.1. Model Building

Whole-body PBPK models for ketoconazole and its metabolites M1 and M2 were developed in PK-Sim^®^ and MoBi^®^. The compiled clinical dataset consisted of 53 studies with a dosing range of 100–1200 mg administered as solutions, capsules, or tablets to 492 participants in total. The respective population characteristics and details of the clinical trials are listed in Appendix A.

As depicted in Figure 1, ketoconazole is metabolized by CYP3A4, AADAC, and UGT1A4. Here, AADAC catalyzes the formation of M1, which is further metabolized by flavin-containing monooxygenase 3 (FMO3) to M2. These processes were implemented via Michaelis–Menten kinetics using Michaelis–Menten constants (K_M_) for AADAC and FMO3 transformations from the literature. As no K_M_ value for CYP3A4 metabolism of ketoconazole was found in the literature, the inhibition constant (K_i_) of ketoconazole’s CYP3A4 inhibition was used as a surrogate value for K_M_ [13]. M2 metabolism via FMO3 was implemented as FMO3-mediated first-order clearance, as no data on this process were available. The developed parent–metabolites model included reversible autoinhibition of CYP3A4 and P-gp. DDIs with the CYP3A4 victim drugs alfentanil, alprazolam, midazolam, and triazolam as well as the P-gp victim drug digoxin were simulated. An overview of the drug-dependent parameters and the respective implemented metabolic processes is summarized in Table 1 and listed in more detail in Appendix A.

### 3.2. Drug–Food Interaction Model Evaluation

The PK of ketoconazole was investigated under both fasted and fed conditions, as clinical data showed considerable influences of food intake on the plasma levels of ketoconazole, especially for doses below 400 mg. To simulate oral solutions, capsules, and tablets, particle dissolution was simulated either with negligible particle radii under 0.002 µm for immediately dissolved particles or with a particle size distribution extrapolated from in vitro data [43] as described in Appendix A. Exemplary simulations of ketoconazole administrations as single and multiple doses under fasted conditions are presented in Figure 2. For this, the observed plasma concentration-time profiles were well described for ketoconazole and its metabolite M1. Model predictions and observations of all plasma concentration-time profiles can be found in Appendix A. 

To model the effect of food intake, the gastric emptying time was optimized to 45 min, which was three times higher than for fasted simulations. Additionally, the intestinal permeability was adapted for fed simulations separately by optimizing the parameter to observed data. Here, the adapted permeability for fed simulations was 1.6-fold lower compared to the fasted state (9.95·10^−6^ versus 1.56·10^−5^ cm/min) as listed in Table 1.

To further underline the impact of DFIs, Figure 3a–c show exemplary plasma concentration-time profiles of ketoconazole administrations under fed conditions, while Figure 3d,e depict comparisons of participants in fasted and fed states. For this, the participants either received ketoconazole after an overnight fast or at the end of a standard breakfast [6,49]. Model-predicted plasma concentration-time profiles were illustrated for doses of 200, 400, and 600 mg during the fasted and fed state alongside their respective observed data. Here, the effect of DFIs was well predicted, especially for the delayed plasma concentrations in fed conditions. Comparisons of observed and predicted plasma concentration-time profiles of 800 mg ketoconazole are shown in Appendix A. Graphical comparisons of all predicted and observed plasma concentration-time profiles are shown on a linear and semi-logarithmic scale in Appendix A.

The general model performance is shown in Figure 4 as the comparison of the predicted and observed AUC_last_ and C_max_ values for the training (a,b) and test dataset (c,d). The PK of ketoconazole and its metabolite M1 was well predicted for fasted, fed, and unknown food states. As shown in Table 2, the overall MRD of 1.45 and the respective GMFEs of 1.37 for AUC_last_ (1.00–2.57) and 1.26 for C_max_ (1.00–2.15) underlined an adequate model performance. Here, 69/77 of the predicted AUC_last_ and the predicted C_max_ values were within the two-fold acceptance limits.

Figure 4e,f depict the predicted compared to observed DFI ratios calculated for AUC_last_ (e) and C_max_ (f). Table 3 lists the mean GMFEs of the predicted compared to observed DFI PK ratios stratified according to the administered dose. For AUC_last_, DFIs were more pronounced for single doses of 400 and 600 mg of ketoconazole and showed an approximately 50% increase in the observed AUC_last_ [6]. For C_max_, the impact of DFIs decreased with increasing doses; the strongest effect was predicted and observed for single doses of 200 mg. Here, C_max_ decreased up to 33% under DFIs [6,49,51]. For the administration of 800 mg, the DFI effect on C_max_ was negligible. With an overall GMFE of 1.19 (1.02–1.47) for AUC_last_ and 1.15 (1.02–1.32) for C_max_, the model predictions for the DFI ratios were in good agreement with the observed data. Here, 7/7 of the AUC_last_ and C_max_ ratios were within the prediction success limits suggested by Guest et al. with a 1.25-fold variability [52]. Implemented DFIs are further documented in Appendix A. Moreover, Appendix A lists the calculated GMFE values of the predicted and observed plasma concentration-time profiles and the corresponding AUC_last_ and C_max_ values along with the respective DFI PK ratios.

Sensitivity analyses for a 7-day multiple-dose simulation of 200 mg of ketoconazole once daily showed that ketoconazole AUC was especially sensitive to changes in the parent’s lipophilicity and fraction unbound. Moreover, changes in the gastric emptying time as well as metabolism via and inhibition of CYP3A4 were among the model parameters to which the AUC was most sensitive. Further details on the performed sensitivity analyses are provided in Appendix A.

### 3.3. Drug–Drug Interaction Modeling and Evaluation

For DDI model performance evaluation, 31 clinical DDI studies covering the CYP3A4 victim drugs alfentanil, alprazolam, midazolam, and triazolam as well as the P-gp victim drug digoxin were used. The collected DDI studies investigated the concomitant treatment of the respective perpetrator alongside the victim drug as well as time-delayed administrations of the perpetrator and victim drugs. Information about the used PBPK models of the victim drugs with the respective model parameters are listed in Appendix A.

First, CYP3A4 and P-gp DDIs with (auto)inhibition by ketoconazole were simulated with a reversible inhibition via the parent compound alone (DDI scenario: P). For this, the respective K_i_ values that described the inhibition of CYP3A4 and P-gp were extracted from the literature [13]. Here, ketoconazole DDIs were simulated for perpetrator and victim drug administration without and with a dosing time gap.

Second, to examine the possible effects of ketoconazole’s metabolites, the DDIs were extended to include reversible inhibition by M1 (DDI scenario: P + M1) and M2 (DDI scenario: P + M1 + M2) as well. While M1 was reported to inhibit CYP3A4 and P-gp and implemented K_i_ values for M1 could be derived from the literature [13], the inhibition by further metabolites was described via inclusion of M2 with K_i_ values for inhibition by M2 surrogated by M1 K_i_ values, as no in vitro data were available (see Table 1).

To investigate the impact of these metabolites (especially for the (long-lasting) DDI potential of ketoconazole), the DDI model performance was compared between simulations with an inhibitory effect by the parent alone (P), the parent with first metabolite (P + M1), and the parent with both metabolites (P + M1 + M2).

Figure 5a demonstrates the predicted concentrations of ketoconazole and its metabolites after a single oral ketoconazole dose in liver cells. The second metabolite (M2) showed a T_max_ roughly 10 h later and a 3.7-fold higher half-life (t_1/2_) in the liver than ketoconazole itself (T_max_: 13.05 vs. 2.20 h; t_1/2_: 45.94 vs. 12.31 h). While the model predicted low concentrations of M1 in plasma (see Figure 2c), thereby indicating minor extracellular distribution, no M2 was simulated to distribute into the plasma.

Figure 5b–e illustrate the plasma concentration-time profiles of the victim drugs given for the case of concomitant dosing with ketoconazole alongside their respective observed data. Here, representative studies of midazolam (Figure 5b,c), alprazolam (Figure 5d), and digoxin (Figure 5e) are depicted and the three DDI scenarios P, P + M1, and P + M1 + M2 compared. In contrast to DDIs with ketoconazole alone (DDI scenario P), model predictions with all three compounds as perpetrators (DDI scenario P + M1 + M2) performed better for the ketoconazole–midazolam and ketoconazole–alprazolam DDIs. Here, the DDI AUC_last_ ratio of the ketoconazole–alprazolam DDI was 0.97 for the DDI scenario P + M1 + M2, while it was only 0.67 for the scenarios P + M1 and P. Here, the DDI scenarios P + M1 and P were very similar, and simulations without dosing time gaps between the victim and perpetrator could not be distinguished by the naked eye. All three scenarios simulated a similar DDI effect for the ketoconazole–digoxin interaction.

Similarly, Figure 6 shows the exemplary plasma concentration-time profiles of the victim drugs given in DDIs with a dosing time gap between victim and perpetrator for alfentanil (Figure 6a,b), midazolam (Figure 6c,d), alprazolam (Figure 6e), and triazolam (Figure 6f) alongside their respective observed data. Again, DDIs were simulated for the three DDI scenarios (P, P + M1, and P + M1 + M2).

Simulations of scenario P performed worst in all illustrated DDIs by underpredicting most of the DDI plasma concentration-time curves. In particular, for the ketoconazole–alfentanil DDIs, predictions with ketoconazole alone showed no effect, as the simulated concentrations were comparable to the respective reference simulation; e.g., alfentanil administration without perpetrator intake. For DDIs with dosing time gaps of 8 h and longer, simulations that included only M1 (DDI scenario P + M1) performed better compared to simulations of ketoconazole alone (DDI scenario P). For example, the DDI AUC_last_ ratios were 0.25 (DDI scenario P + M1) and 0.18 (DDI scenario P) for the ketoconazole–alfentanil DDIs (see Figure 6b). For the remaining simulations, the performance of scenarios P and P + M1 was comparable (as shown in Figure 6d–f).

Overall, the joint parent–metabolites DDI model (DDI scenario P + M1 + M2) demonstrated the most convincing performance compared to the models that included either one or no metabolite (DDI scenarios P + M1 and P) for the prediction of long-lasting DDI effects, especially if ketoconazole was administered several hours before the victim drug.

All simulated DDI profiles with their respective observed data are shown in Appendix A together with a detailed description of regimens and population characteristics in Appendix A.

Figure 7 illustrates the comparison of the predicted and observed DDI ratios for the AUC_last_ and C_max_ of all victim drugs. Figure 7a–c show the calculated DDI AUC_last_ ratios, whereas Figure 7c–e depict the respective calculated DDI C_max_ ratios. Goodness-of-fit plots were stratified for the three DDI scenarios (P, P + M1, and P + M1 + M2).

Here, 26/27 of the predicted DDI AUC_last_ ratios of the P + M1 + M2 DDI model were within the limits proposed by Guest et al. [52], while only 12/27 of the DDI ratios were well predicted for the P and P + M1 DDI models. All predicted DDI AUC_last_ ratios for DDI simulations with dosing time gaps between victim and perpetrator administration were outside of the acceptance limits. For DDI C_max_ predictions, all 21/21 of the DDI ratios of the joint P + M1 + M2 model were within the limits proposed by Guest et al. [52], while only 19/21 and 18/21 met the acceptance criterion if DDIs were simulated for P + M1 and P, respectively.

The mean GMFE values of the calculated DDI PK ratios of all victim drugs are shown in Figure 7g,h for existing dosing time gaps stratified according to the three DDI scenarios (P, P + M1, and P + M1 + M2).

The overall GMFEs for the DDI performance that included both metabolites (P + M1 + M2) were 1.35 (1.01–2.41) for DDI AUC_last_ and 1.27 (1.02–1.96) for DDI C_max_. For the DDI model that included only M1 (P + M1), the mean GMFEs for DDI AUC_last_ and DDI C_max_ were 2.44 (1.01–5.34) and 1.42 (1.02–3.28), respectively. For DDI prediction without metabolite inhibition (P), the mean GMFEs were 2.64 (1.01–6.75) for DDI AUC_last_ and 1.52 (1.02–4.15) for DDI C_max_. In general, the DDI model performance was the best for DDI P + M1 + M2 compared to DDI P + M1 and DDI P, and there was a larger impact on DDI AUC_last_ than on DDI C_max_ ratios. The calculated AUC_last_ and C_max_ ratios as well as the GMFE values of all predicted DDI studies are listed in Appendix A Appendix A.

## 4. Discussion

A whole-body PBPK model for ketoconazole and its metabolites M1 and M2 was built and evaluated to cover ketoconazole administrations as oral solutions, capsules, or tablets for a wide dosing range of 100–1200 mg to model DFIs and CYP3A4 and P-gp DDIs.

The available literature lacked studies on ketoconazole intravenous injections or infusions in humans, and only data on oral or dermal applications were available [3]. For oral intake, the absorption of ketoconazole is highly limited by its poor solubility, which rapidly decreases with increasing pH [4]. As food consumption can influence gastric pH, it is reasonable to assume that this might also modulate the oral bioavailability of ketoconazole [6]. The liberation of oral ketoconazole formulations was described as a particle-dissolution process. For oral solutions, particles were assumed to be immediately dissolved; in the case of oral tablets, particle radii and distribution were estimated from observed data [57]. Moreover, supersaturation of the poorly soluble ketoconazole over the modeled dosing range (up to 1200 mg) was assumed, since in the current literature, potential oversaturation was discussed for ketoconazole and other imidazole derivatives with known poor solubility [58].

Generally, the intake of food might lead to delayed gastric emptying times of up to two hours depending on the meal composition [38]. Thus, ketoconazole’s residence time in the gut (and therefore at the absorption site) can be prolonged during DFIs. This can result in an increased absorption as well as a delay in T_max_. To model ketoconazole absorption in the fed state, a specific intestinal permeability was estimated, and the transit time in the stomach compartment was prolonged to describe the delay in T_max_ compared to the fasted state. Here, the gastric emptying time was set to 15 min (default value) for fasted simulations. For all fed simulations, a gastric emptying time of 45 min was optimal to describe the observed data, although the relative time of food intake varied in the investigated studies; i.e., either simultaneous intake or 0.5–1 h before or after ketoconazole administration. For doses of 200–600 mg, T_max_ was delayed by 1–1.5 h compared to ketoconazole administered in a fasted state [6]. Observed C_max_ values were not affected by food intake, while the respective AUC_last_ values were higher for fasted scenarios [6]. With increasing doses of administration, differences in plasma exposure were less pronounced for ketoconazole. For AUC_last_, the impact of DFIs was especially relevant for doses of 400 and 600 mg with observed DFI ratios of 1.59 and 1.45, respectively, while only unnoticeable differences in the plasma concentration-time curves between 800 mg ketoconazole in the fasted and fed states could be observed [6].

If the intake of food was not specified in the clinical study protocol, a DFI was assumed if fasted simulations were not appropriate to describe the respective data. This was the case in the following scenarios: First, if T_max_ was observed more than two hours after ketoconazole administration. Second, if multiple doses were administered within a day or over several days, as it was assumed that participants were not in a fasted state throughout the entirety of their study protocol. Third, if doses of orally administered ketoconazole were higher than 600 mg, as differences in absorption between the fasted and fed states substantially decrease with increasing dose (e.g., differences in the observed T_max_ and C_max_ of 800 mg ketoconazole were unnoticeable in both cases) [6]. This can be explained by a delayed absorption of higher doses due to ketoconazole’s limited solubility rather than the intake of food alone.

The developed ketoconazole PBPK model included metabolism via AADAC and UGT1A4 [7,10]. Here, K_M_ values could be extracted from the literature. Implementation of AADAC-mediated metabolism was essential to describe the formation of M1 and M2. UGT1A4 was implemented to cover ketoconazole degradation irrespective of inhibitory DDI effects, as it was neither involved in the formation of the modeled metabolites nor affected by ketoconazole’s autoinhibition.

Moreover, ketoconazole metabolism via CYP3A4 and transport via P-gp also were implemented as discussed in the literature [8,9,59]. Although CYP3A4-mediated metabolism and P-gp-mediated transport have not been fully investigated for ketoconazole and no information guiding kinetic parametrization (e.g., K_M_ or V_max_) has been reported yet, metabolism via CYP3A4 was implemented to describe a potential accumulation of ketoconazole during multiple-dose administrations due to its autoinhibition [5]. The K_i_ that described the CYP3A4 autoinhibition by ketoconazole was used as a surrogate value for a missing K_M_ of CYP3A4 since a similar parametrization strategy was already successfully applied during the PBPK model development of the imidazole derivative itraconazole by Hanke et al. [22]. The implementation of P-gp as an efflux transporter was included to thoroughly describe ketoconazole excretion, as between 10–37% of unchanged ketoconazole could be found in feces [5]. For this, the presented model predicted a fraction excreted to feces of around 27% after a single-dose administration of 200 mg of ketoconazole as an oral tablet in the fasted state. The K_M_ for P-gp transport was also taken from the K_i_ value used to describe the autoinhibition of P-gp.

The metabolite M1, which is formed by AADAC transformation of ketoconazole, is further metabolized to M2 via FMO3 [7,39]. The K_M_ value of FMO3-mediated metabolism was derived from the literature [39], while k_cat_ was optimized to fit the observed data. However, only one study by Weiss et al. reported plasma concentration-time profiles of M1 [13]; its exposure in plasma was only a fraction of its parent with observed C_max_ values of 6.07 ng/mL compared to 4956.03 ng/mL for ketoconazole after a single dose of 400 mg in the fasted state [13]. As metabolite concentrations are low in plasma, the authors assumed M1 accumulation in the liver [13]. In previous studies, M1 could not even be detected in plasma [60], and its metabolite M2, which is also metabolized by FMO3, was never reported to appear in plasma. Thus, in the present parent–metabolites PBPK model, M2 accumulation in liver cells was assumed, and permeation into plasma was prevented to account for the lack of reported M2 quantification in plasma. However, to precisely assess M2 disposition kinetics, more research is required. For the FMO3-mediated metabolism of M2 in the cell [39] no in vitro measurements were available. Hence, an FMO3-mediated clearance was implemented and optimized to thoroughly predict the respective DDIs. It should be noted that the implementation of M2-mediated inhibition captured the potential involvement of several metabolites, which could be important to ketoconazole’s inhibitory effect. Here, M2 served as a representative metabolite of various metabolites that are not fully understood and also need to be further investigated [21,39].

While M1 has also been observed to inhibit CYP3A4 and P-gp [7,13] (with respective K_i_ values available in the literature), M2-mediated inhibition was not reported. To investigate the importance of both M1 and M2, which may account for further unknown metabolites, for ketoconazole’s inhibitory effect, DDIs were simulated for the parent alone (P), with only M1 (P + M1), and with both metabolites (P + M1 + M2). For M2-mediated inhibition of CYP3A4 and P-gp, the respective K_i_ values were surrogated from literature K_i_ values used for M1 inhibition. CYP3A4 and P-gp DDIs were simulated with the victim drugs alfentanil, alprazolam, midazolam, triazolam, and digoxin.

In general, the modeling of both metabolites (P + M1 + M2) outperformed predictions without metabolites (scenarios P and P + M1), and apparent differences were most notable in the following two scenarios: First, if victim drug exposure was observed over a long time. Here, victim drug plasma concentrations measured 10 hours after administration were better predicted if modeling of M1 and M2 was included (Figure 5). Second, DDI performance of P + M1 + M2 was superior if the perpetrator and victim drug were administered at different times. This was especially apparent when comparing the DDI AUC_last_ and C_max_ ratios in evaluations without modeling M1 and M2. In the case of dosing time gaps between the victim and perpetrator, the inclusion of only M1 performed slightly better than modeling the DDIs with only the parent compound. For the remaining model scenarios, P + M1 and P performed equally well. Moreover, the modeling of M1 or M2 did not impact ketoconazole exposure.

Simulations of various DDI scenarios illustrated that reversible inhibition via ketoconazole alone was not sufficient to describe the impact on AUC_last_ and C_max_ of the victim compounds, especially if dosing time gaps of several hours between the victim and perpetrator were considered. Similarly, a successfully developed parent–metabolites PBPK model for itraconazole included its three metabolites that participated in the DDIs as well [22]. Hence, it is reasonable to assume that ketoconazole is not solely responsible for its DDI effect. For ketoconazole, a potential mechanism-based inhibition; e.g., of CYP3A4, was discussed, but the contributions of ketoconazole’s metabolites were not investigated [61,62]. In more recent studies, a reversible rather than a mechanism-based inhibition was reported [13,63]. In the present work, all inhibitions were described as reversible inhibitions, as it could be assumed that the supposed mechanism-based inhibition of ketoconazole might be a sequence of reversible inhibitions by ketoconazole and its metabolites intracellularly.

Overall, the developed parent–metabolites PBPK model of ketoconazole was capable of describing and predicting DDIs with CYP3A4 and P-gp victims successfully, especially for dosing time gaps between perpetrator and victim drug administration. There were potential biases when it came to the model development and application. Potential sources of bias might have included: (i) the selection of clinical study reports for the training and test datasets from publicly available data sources; (ii) the demographic distribution of modeled individuals due to the inclusion criteria of the respective clinical trials and thus potential heterogeneities in the respective physiology of the investigated participants; (iii) heterogeneities in the modeled pathophysiology (mostly healthy individuals were covered in our analysis); and (iv) the potential to miss the implementation of important but yet unknown metabolism or transport processes. Moreover, several assumptions had to be made to inform the implemented processes; for example, assuming K_M_ values to estimate k_cat_ values for CYP3A4 metabolism or P-gp transport. A previously published PBPK model of ketoconazole discussed its impact as a perpetrator on DDIs with alprazolam and midazolam [15]. For this, only ketoconazole administrations of 200 mg twice daily and 400 mg once daily as oral solutions in the fed state were investigated. In addition, our modeling work also investigated the inhibition of P-gp and the potential involvement of ketoconazole metabolites in DDIs to cover a broad dosing regimen with multiple victim drugs. Based on the presented simulations, the modeled metabolites might play a crucial and important role in the overall inhibitory effect of ketoconazole. The developed PBPK models can serve to generate hypotheses regarding the impact of metabolites on a drug’s interaction potential, especially in polymedicated individuals. Moreover, since ketoconazole is classified by the FDA as a strong inhibitor of CYP3A4 and P-gp, the presented PBPK models can be coupled with further victim models to simulate different DDI scenarios and also to interpret the extensive DDI evidence already collected using this compound.

## 5. Conclusions

A parent–metabolites PBPK model for ketoconazole and its metabolites M1 and M2 was developed. The comprehensive PBPK model was capable of predicting the effect of DFIs on ketoconazole. Moreover, the presented model captured the potential importance of metabolites for ketoconazole’s prominent inhibitory effect as a CYP3A4 and P-gp perpetrator drug in various investigated DDI scenarios. The PBPK model files are freely available at http://models.clinicalpharmacy.me to support further DDI studies in drug development and discovery.

## Figures and Tables

**Figure 1 pharmaceutics-15-00679-f001:**
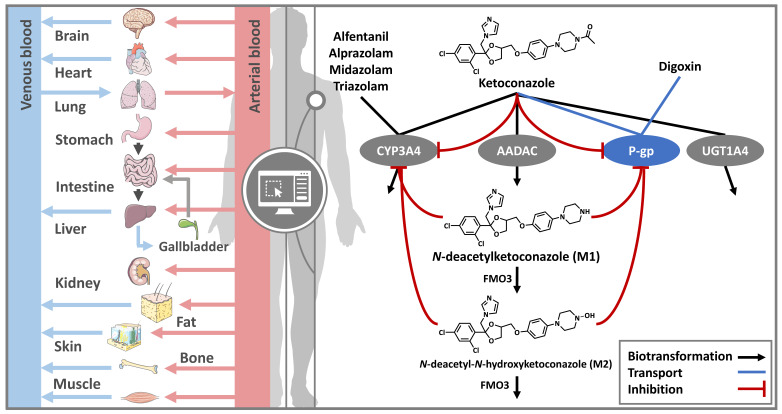
DDI PBPK modeling overview. Whole-body PBPK models for ketoconazole and its metabolites were established and used to simulate the inhibitory effect of ketoconazole, which is substrate of CYP3A4, AADAC, UGT1A4, and P-gp. Its metabolite M1, which is formed by AADAC biotransformation, is metabolized by FMO3 to M2, which is metabolized via FMO3 as well. Both the parent compound and the metabolites concomitantly inhibit CYP3A4 and P-gp. CYP3A4-related DDIs were simulated with the CYP3A4 victim drugs alfentanil, alprazolam, midazolam, and triazolam. P-gp DDIs were simulated with the P-gp victim drug digoxin. Drawings by Servier (licensed under CC BY 3.0) [32]. AADAC: arylacetamide deacetylase, CYP3A4: cytochrome P450 3A4, FMO3: flavin-containing monooxygenase 3, M1: *N*-deacetylketoconazole, M2: *N*-deacetyl-*N*-hydroxyketoconazole, P-gp: P-glycoprotein, UGT1A4: uridine diphosphate glucuronosyltransferase 1A4.

**Figure 2 pharmaceutics-15-00679-f002:**
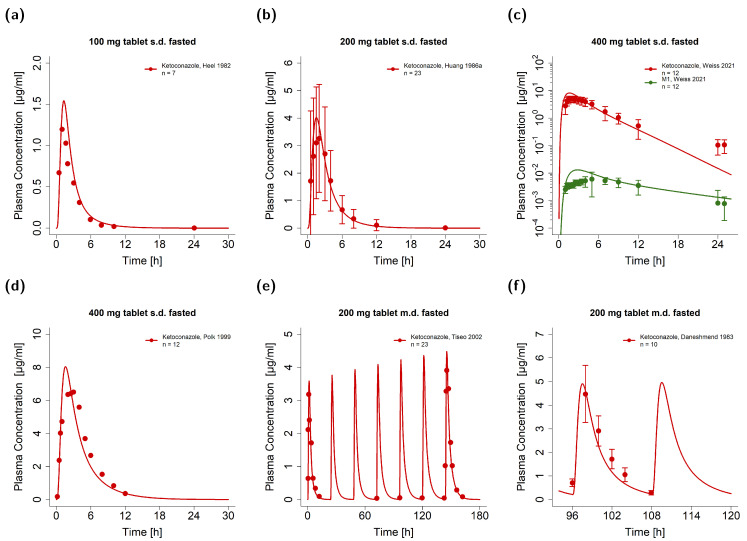
Graphical comparison of predicted and observed plasma concentration-time profiles of exemplary clinical trials of ketoconazole under fasted and fed conditions. (**a**–**d**) Single-dose administrations of tablets in fasted state with metabolite measurements; (**e**,**f**) multiple-dose administrations of capsules and tablets in fasted state [1,13,44,45,46,47,48]. The model predictions are shown as solid lines and the corresponding observed data as dots (arithmetic mean ± standard deviation (if available)). Detailed information on study protocols is provided in Appendix A. fasted: fasted condition, fed: fed conditions, M1: *N*-deacetylketoconazole, n: number of study participants.

**Figure 3 pharmaceutics-15-00679-f003:**
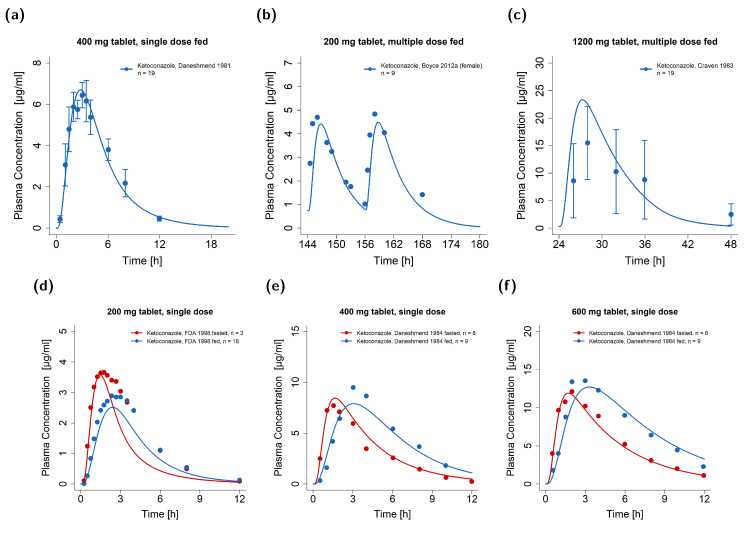
Ketoconazole DFI model performance. Illustrated are plasma concentration-time profiles of exemplary clinical trials of (**a**–**c**) single- and multiple-dose administrations of tablets in fed state [46,48,50]. Moreover, comparisons of fasted (red) and fed (blue) predicted and observed plasma concentration-time profiles are illustrated for 200 mg (**d**), 400 mg (**e**), and 600 mg (**f**) single-dose administrations of ketoconazole [6,49]. The model predictions are shown as solid lines and the corresponding observed data as dots (arithmetic mean). Detailed information on the study protocols is provided in Appendix A. fasted: fasted condition; fed: fed condition; n: number of participants.

**Figure 4 pharmaceutics-15-00679-f004:**
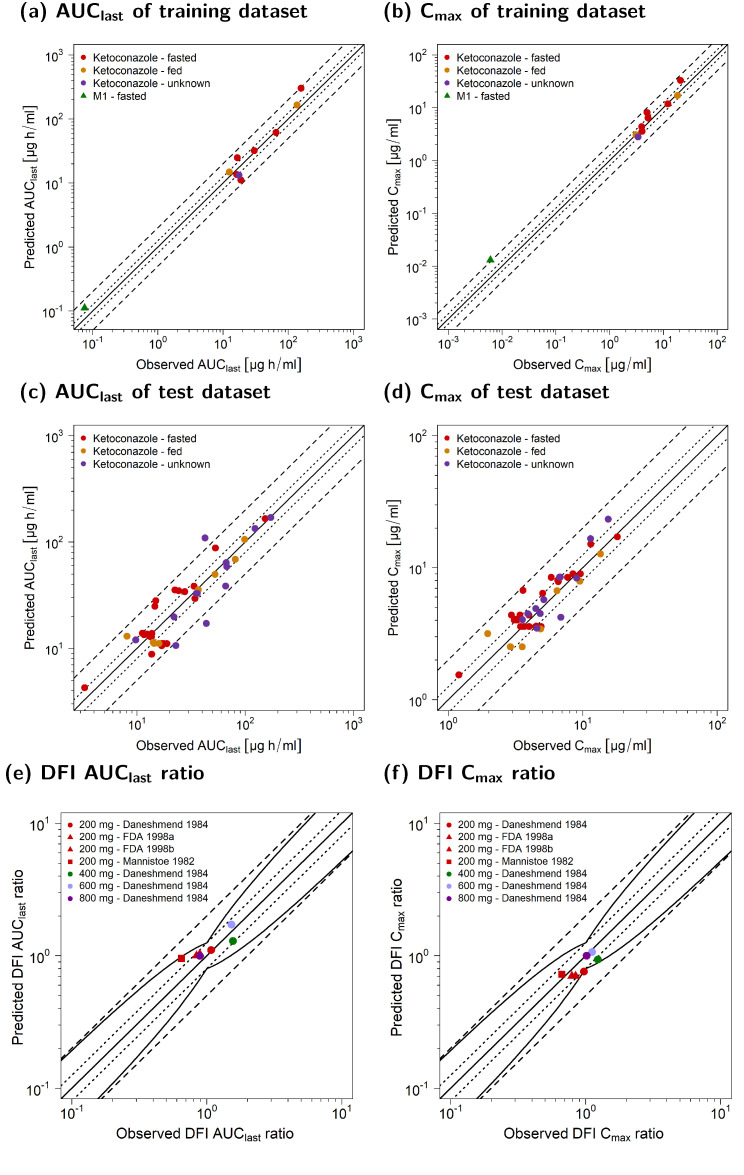
Goodness-of-fit plots of PK parameters for ketoconazole and M1. Predicted AUC_last_ values of the training (**a**) and test dataset (**c**) as well as C_max_ values of the training (**b**) and test dataset (**d**) were compared to the respective observed data. Predicted (as compared to observed) DFI effect ratios of AUC_last_ (**e**) and C_max_ (**f**) are shown for the single doses of 200, 400, 600, and 800 mg [6,49,51]. The straight solid line marks the line of identity, dotted lines indicate 1.25-fold, and dashed lines indicate 2-fold deviation. The curved solid lines show the prediction acceptance limits proposed by Guest et al. (including 1.25-fold variability) [52]. Detailed information on the study protocols is provided in Appendix A. AUC_last_: area under the plasma concentration-time curve calculated from the first to the last concentration measurement; C_max_: maximum plasma concentration; DFI: drug–food interaction; fasted: fasted condition; fed: fed condition, M1: *N*-deacetylketoconazole; n: number of study participants; unknown: unknown food intake.

**Figure 5 pharmaceutics-15-00679-f005:**
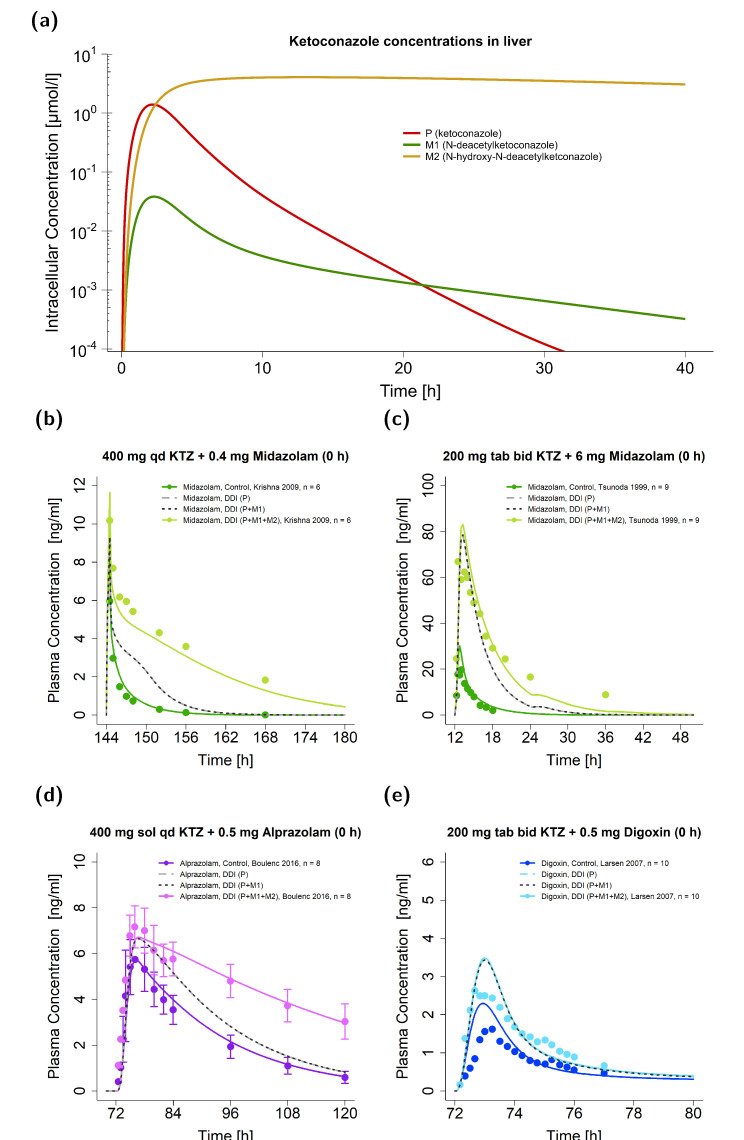
Ketoconazole DDI model simulations without a dosing time gap between victim and perpetrator. Predicted intracellular concentrations in liver cells are illustrated for ketoconazole and its metabolites (M1 and M2) on a semi-logarithmic scale (**a**). Predicted compared to observed plasma concentration-time profiles are illustrated for DDIs with the victim drugs midazolam (**b**,**c**), alprazolam (**d**), and digoxin (**e**) [15,53,54,55]. Illustrated are DDI predictions (i) with the parent alone (P) (long dashed line in grey), (ii) with the parent and M1 (P + M1) (dashed line in black), and (iii) with the parent and both metabolites (P + M1 + M2) (solid line in a brighter colored shade) alongside their respective reference profile (solid line in a darker colored shade). Corresponding observed data are shown as dots (arithmetic mean ± standard deviation (if available)). Detailed information on the study protocols is provided in Appendix A. bid: twice daily; DDI: drug–drug interaction; KTZ: ketoconazole; M1: *N*-deacetylketoconazole; M2: *N*-hydroxy-*N*-deacetylketoconazole; n: number of participants; P: ketoconazole alone; qd: once daily; sd: single dose, sol: solution; tab: tablet.

**Figure 6 pharmaceutics-15-00679-f006:**
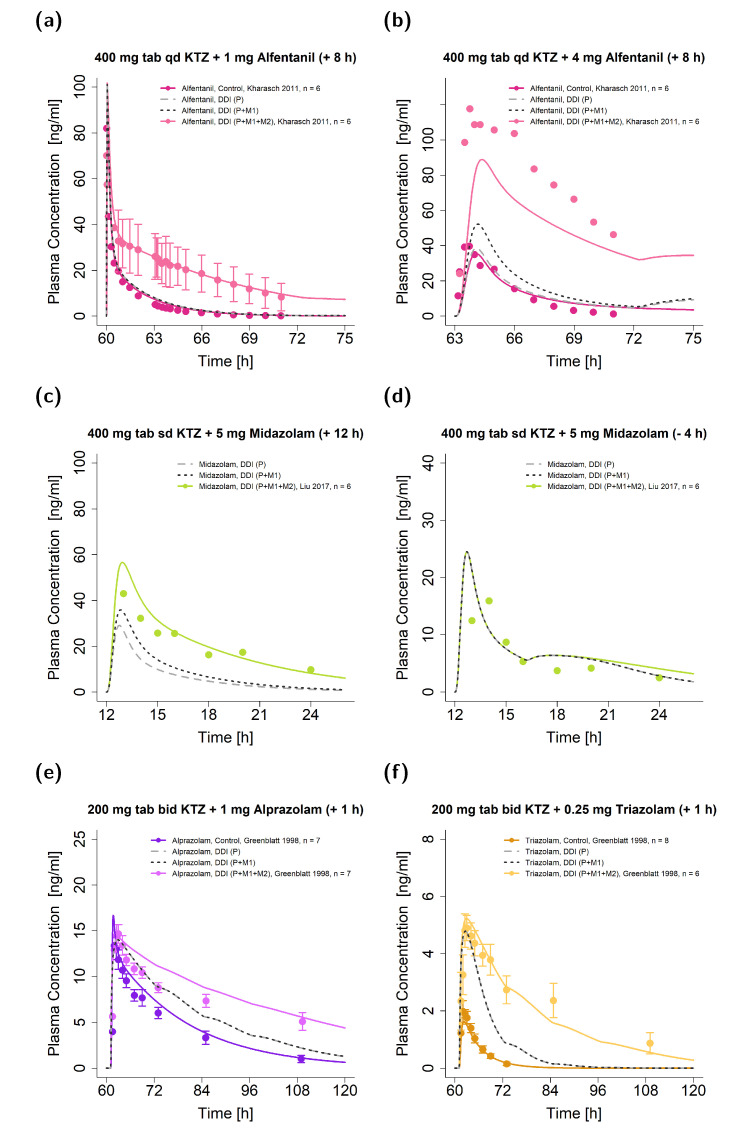
Ketoconazole DDI model simulations. Predicted (as compared to observed) plasma concentration-time profiles are illustrated for DDIs with the victim drugs alfentanil (**a**,**b**), midazolam (**c**,**d**), alprazolam (**e**), and triazolam (**f**) [17,19,56]. The time of victim drug intake was 8 or 12 h after (**a**–**c**), 4 h before (**d**), and 1 h after (**e**,**f**) ketoconazole administration. Illustrated are DDI predictions (i) with the parent compound alone (P) (long dashed line in grey), (ii) with the parent compound and M1 (P + M1) (dashed line in black), and (iii) with the parent compound and both metabolites (P + M1 + M2) (solid line in a brighter colored shade) alongside their respective reference profiles (solid line in a darker colored shade). Corresponding observed data are shown as dots (arithmetic mean ± standard deviation (if available)). The dosing of ketoconazole–alfentanil DDIs (**a**,**b**) was normalized to the respective control to highlight the comparison of DDI and control, while the Appendix A show the data from the respective studies that were simulated as described in their clinical trials reports for the DDI model evaluation and documentation [19]. Detailed information on the study protocols is provided in Appendix A. For the DDI studies illustrated in (**c**,**d**), no reference profiles were available. Note: bid: twice daily; DDI: drug–drug interaction; KTZ: ketoconazole; M1: *N*-deacetylketoconazole; M2: *N*-hydroxy-*N*-deacetylketoconazole; n: number of participants; P: ketoconazole alone; qd: once daily; sd: single dose; tab: tablet.

**Figure 7 pharmaceutics-15-00679-f007:**
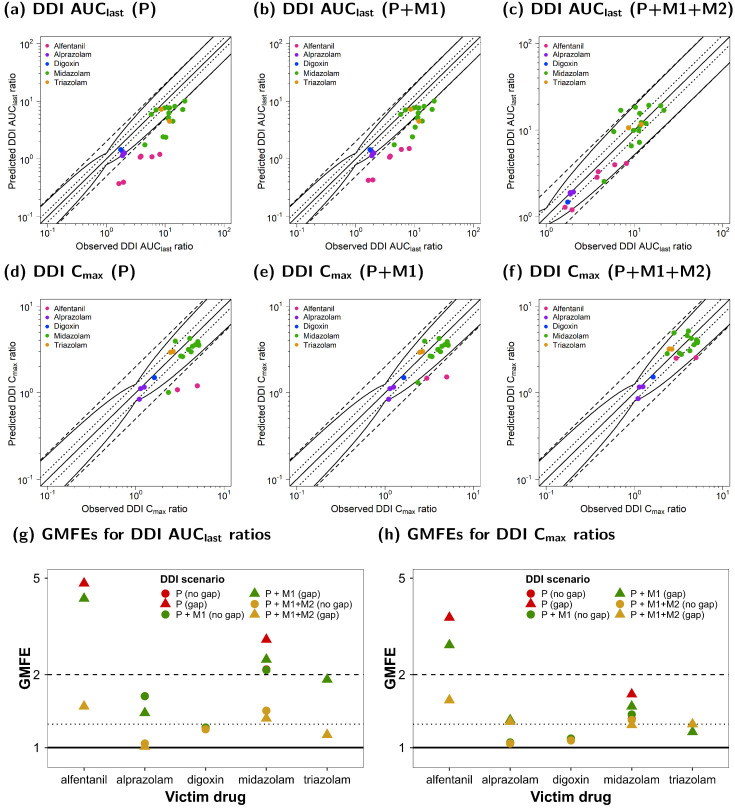
Ketoconazole DDI model evaluation. Predicted DDI AUC_last_ ratios of DDI simulations of three scenarios (P (**a**), P + M1 (**b**), and P + M1 + M2 (**c**)), as well as DDI C_max_ ratios of three scenarios (P (**d**), P + M1 (**e**), and P + M1 + M2 (**f**)) were compared to the respective observed data. The straight solid line marks the line of identity, and the curved solid lines show the prediction acceptance limits proposed by Guest et al. (including 1.25-fold variability) [52]. Calculated mean GMFE values for DDI AUC_last_ (**g**) and DDI C_max_ ratios (**h**) for the three scenarios (P, P + M1, and P + M1 + M2) stratified according to victim with or without a dosing time gap between ketoconazole administration. Dotted lines indicate 1.25-fold and dashed lines indicate two-fold deviation. Detailed information on the study protocols is provided in Appendix A. AUC_last_: area under the plasma concentration-time curve calculated from the first to the last concentration measurement; C_max_: maximum plasma concentration; DDI: drug–drug interaction; GMFE: geometric mean fold error; KTZ: ketoconazole; M1: *N*-deacetylketoconazole; M2: *N*-hydroxy-*N*-deacetylketoconazole; P: ketoconazole alone.

**Table 1 pharmaceutics-15-00679-t001:** Drug-dependent parameters of the parent–metabolite PBPK models for ketoconazole, M1 and M2.

Parameter	Unit	Value	Literature	Value	Literature	Value	Literature	Reference	Description
		Ketoconazole		M1		M2			
MW	g/mol	531.43	531.43	489.40	489.40	505.40	505.40	[33,34,35]	Molecular weight
pKa	-	2.94 6.51	2.94 6.51	0.20 6.42 8.90	0.20 6.42 8.90	3.42 6.42	3.42 6.42	[33,34,35]	Acid dissociation constant
Solubility (pH)	mg/L	2.03·10^4^ (1.2) 4.3·10^4^ (3) 7.00 (6.8). 5.40 (7), 6.00 (7.5)	2.03·10^4^ (1.2) 4.3·10^4^ (3) 7.00 (6.8). 5.40 (7), 6.00 (7.5)	1.24·10^3^ (6.5)	1.24·10^3^ (6.5)	4.40·10^3^ (6.5)	4.40·10^3^ (6.5)	[4,34,35]	Solubility
log P	-	^o^ 2.52	2.73	^o^ 3.75	4.58	4.20	4.20	[34,35,36]	Lipophilicity
fu	%	1.00	1.00	^a^ 1.00	-	^a^ 1.00	-	[1]	Fraction unbound
Partition coefficients	-	Various	Berez.	Various	R&R	Various	Berez.	-	Cell-to-plasma partitioning
Cellular perm.	-	-	PK-Sim	-	Ch.d.S.	-	Ch.d.S.		Permeability into the cellular space
GFR fraction	-	1.00	-	1.00	-	1.00	-	-	Fraction of filtered drug in urine
EHC cont. fraction	-	1.00	-	1.00	-	1.00	-	-	Bile fraction continuously released
Intest. perm. fasted	cm/min	^o^ 1.56·10^−5^	^c^ 4.28·10^−6^	-	^-^	-	^-^	[37]	Transcellular intestinal permeability
Intest. perm. fed	cm/min	^o^ 9.95·10^−6^	^c^ 4.28·10^−6^	-	^-^	-	^-^	[37]	Transcellular intestinal permeability
GET fasted	min	15	^d^ 15	-	-	-	-	[37]	Gastric emptying time
GET fed	min	^a^ 45	45–120	-	-	-	-	[38]	Gastric emptying time
K_M_ AADAC	µmol/L	1.88	1.88	-	-	-	-	[7]	Michaelis–Menten constant
k_cat_ AADAC	1/min	^o^ 0.87	-	-	-	-	-	-	Catalytic rate constant
K_M_ CYP34	µmol/L	^a^ 0.008	-	-	-	-	-	-	Michaelis–Menten constant
k_cat_ CYP34	1/min	^o^ 0.10	-	-	-	-	-	-	Catalytic rate constant
K_M_ UGT1A4	µmol/L	7.00	7.00	-	-	-	-	[10]	Michaelis–Menten constant
k_cat_ UGT1A4	1/min	^o^ 0.31	-	-	-	-	-	-	Catalytic rate constant
K_M_ FMO3	µmol/L	-	-	1.77	1.77	-	-	[39]	Michaelis–Menten constant
k_cat_ FMO3	1/min	-	-	^o^ 378.65	-	-	-	-	Catalytic rate constant
Cl FMO3	l/µmol/min	-	-	^-^	-	^o^ 0.09	-	-	First order clearance rate constant
K_M_ P-gp	µmol/L	^a^ 0.035	-	-	-	-	-	-	Michaelis–Menten constant
k_cat_ P-gp	1/min	^o^ 0.33	-	-	-	-	-	-	Catalytic rate constant
K_i_ CYP3A4	µmol/L	0.008	^‡^ 0.008	0.022	^‡^ 0.022	^a^ 0.022	-	[13]	Conc. for half-maximal inhibition
K_i_ P-gp	µmol/L	0.035	^‡^ 0.035	0.119	^‡^ 0.119	^a^ 0.119	-	[13]	Conc. for half-maximal inhibition

^‡^ In vitro values calculated from respective IC50 values. ^a^ Assumed; ^c^ calculated; ^d^ default value; ^o^ optimized value. AADAC: arylacetamide deacetylase; Berez.: Berezhkovskiy calculation method [40]; Ch.d.S.: charge-dependent Schmitt calculation method [41]; conc.: concentration; cont.: continuous; CYP3A4: cytochrome P450 3A4; EHC: enterohepatic circulation; FMO3: flavin-containing monooxygenase 3; GET: gastric emptying time; GFR: glomerular filtration rate; intest.: intestinal; KTZ: ketoconazole; M1: *N*-deacetylketoconazole; M2: *N*-deacetyl-*N*-hydroxylketoconazole; perm.: permeability; P-gp: P-glycoprotein; PK-Sim: PK-Sim^®^ standard calculation method [37]; R&R: Rodgers and Rowland calculation method [42]; UGT1A4: uridine diphosphate glucuronosyltransferase 1A4.

**Table 2 pharmaceutics-15-00679-t002:** Summary of quantitative measures of model performance for ketoconazole and its metabolite M1.

Compound (n)	Mean MRD	Mean GMFE AUC_last_	Mean GMFE C_max_
Ketoconazole (52)	1.42	1.37	1.24
M1 (1)	2.51	1.48	2.15
Overall	1.45	1.37	1.26
Profiles with measure ≤ 2	49/53	50/53	52/53
Range	1.09–2.69	1.00–2.57	1.00–2.15

AUC_last_: area under the plasma concentration-time curve from the time of drug administration to the time of the last concentration measurement; C_max_: maximum plasma concentration; GMFE: geometric mean fold error; M1: *N*-deacetylketoconazole; MRD: mean relative deviation; n: number of mean plasma concentration-time profiles.

**Table 3 pharmaceutics-15-00679-t003:** Predicted and observed DFI PK ratios alongside quantitative measures of DFI model performance.

Single-Dose Ketoconazole (mg) (n)	Mean GMFE DFI AUC_last_	Mean GMFE DFI C_max_
200 (4)	1.21	1.12
400 (1)	1.20	1.23
600 (1)	1.13	1.12
800 (1)	1.13	1.02
Overall GMFE	1.19	1.15
DFIs within guest limits	7/7	7/7
Range	1.02–1.47	1.02–1.32

AUC_last_: area under the plasma concentration-time curve from the time of drug administration to the time of the last concentration measurement; C_max_: maximum plasma concentration; DFI: drug–food interaction; GMFE: geometric mean fold error; n: number of DFI ratios.

## Data Availability

All modeling files (including the clinical study data utilized) can be found at http://models.clinicalpharmacy.me.

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
