# Peer review of "A Physiologically Based Pharmacokinetic Model of Ketoconazole and Its Metabolites as Drug–Drug Interaction Perpetrators"

_pharmaceutics, 2023, doi:10.3390/pharmaceutics15020679_

Round 1

Reviewer 1 Report

Fatima Zahra Marok et al. develop a whole-body physiologically based pharmacokinetic (PBPK) model of ketoconazole and its metabolites for fasted and fed states and to investigate the impact of ketoconazole and its metabolites on its DDI potential. In general, the modelling was well conducted and is in accordance with the state-of-the art. However, I have several major and minor comments that needs to be appropriately addressed before publication.

 Abstract

Line 22/27/28/29: it is unclear what '7/7', '26/27','18/27' ... means?

Introduction

The PK characteristics including absorption, distribution, metabolism, and elimination of ketoconazole needs to be briefly described. Moreover, why systemic administration of ketoconazole for the treatment of fungal infections is not recommended but can used for the Cushing’s syndrome. Why the liver injury, prolonged QT and other safety issues are not concern for the therapy of Cushing’s syndrome.

 The author could concise the introduction section by focusing on what new information will be generated and how it will add to existing knowledge. e.g. The introduction for the itraconazole may not be necessary.

Method & Results

The rationale for the selected drugs (alfentanil, alprazolam, midazolam, triazolam and the P-gp victim drug digoxin) to develop drug-drug interaction PBPK model needs to be presented. Moreover, why a wide dosing range of 100—1200 mg was investigated, not 200-400 mg commonly used in the therapy of Cushing’s syndrome.

 The application for the established PBPK model was not presented.

Discussion

The discussion section should focus on their new findings and not necessary to repeat the description already presented in the other section e.g. line 455-459。

Study limitations describing potential sources of bias and imprecision where relevant should be described.

The clinical relevance of study findings also needs to be describe.

Reviewer 2 Report

This is a typical PBPK work in order to demonstrate the significance and importance of PBPK modeling. The work does not add anything new from PBPK perspective because there are hundreds of drugs which have been used to demonstrate the feasibility of PBPK models for drug-drug interaction, food-drug effect, and the metabolic profiles of drugs. All the characteristics of ketoconazole described here are well known from experimental data and a PBPK model is not needed to learn these characteristics of ketoconazole.  With the advent of minimum PBPK models, the authors would have done a better job by using a minimal PBPK model rather than a whole body PBPK model. This would have not only added some practical value to this work but some innovation.  Based on the literature, it is apparent that whole body PBPK does not provide any better results than minimal PBPK and has become more of a commercial interest rather than scientific interest.

Some comments related to this PBPK modeling exercise are mentioned below.

According to drugbank (https://go.drugbank.com/drugs/DB01212) the logp value of ketoconazole is 4.35 and "ketoconazole is approximately 84% bound to plasma albumin with another 15% associated with blood cells for a total of 99% binding within the plasma". Therefore, unbound fraction of ketoconazole is 0.16 not 1 as authors have used.

The authors used a logP value of 2.52 rather than 4.35 and unbound fraction as 1% (0.001). fup should be 0.16 or 16% ( for fup binding with blood cells should not be accounted for). This is just a comment because changing the values have no impact on the predictive power of the PBPK model.

Optimization is a typical practice in PBPK modeling which leads to an improved prediction of the desired parameter(s) under a given condition. This work states the following:

Page #3, lines 125-128. Model parameters that either could not be adequately informed from the literature or were involved in important QSAR model estimates of permeability and distribution  processes were optimized by fitting the model simultaneously to all plasma concentration-time profiles of the training dataset.

In other words, one needs observed concentration-time data to come up with PBPK model parameters. No such data no PBPK model or a very erratic prediction!

Page #3, lines 133-135. To simulate the effect of  DFIs on oral ketoconazole absorption, intestinal permeabilities, and gastric emptying times were estimated based on the fasted and fed datasets.

Especially, I will question the optimization of ketoconazole absorption, intestinal permeabilities, and gastric emptying times based on fasted and fed data. The objective is to evaluate the impact of food on the PK of ketoconazole and if one does not have data from food study then how one will optimize the aforementioned parameters of ketoconazole?  This kind of optimization based on observed data is bound to give good results. 

Overall, there is nothing new or innovative in this work. Optimization with real data has led to a reasonably good prediction of the desired parameters and the methodology here is nothing new from other PBPK modeling exercise, mainly optimization.

Reviewer 3 Report

In the manuscript entitled “A Physiologically Based Pharmacokinetic Model of Ketoconazole and its Metabolites as Drug-Drug Interaction Perpetrators”, the authors use the software PK-Sim® and MoBi® to quantitatively describe the plasma concentration profile of ketoconazole under fasted and fed conditions, as well as on the effects of Ketoconazole on the plasma concentration profile of Alfentanil, Alprazolam, Midazolam, Triazolam, and Digoxin. Kinetic modeling is a very important tool that may provide a quantitative interpretation of in vivo results, providing insight on the details of what is occurring in the complex in vivo system. To be relevant and useful, the models must be described fully and in detail. This is not unfortunately the case for the work reported in this manuscript. Some of the reviewer concerns regarding the model definition, parameters used, and model results are indicated below.
1 – Poor definition of the Model Parameters. The model is defined regarding the compartments considered and their connectivity (Figure 1) but not regarding the volumes of the distinct compartments nor the rate of transport between the compartments. Several kinetic parameters required for the calculation of the plasma concentration profiles indicated in the figures are also not defined; eg. the rate of dissolution (on line 231 this is incorrectly directed to Section S1.4 where nothing is said about dissolution), degradation, elimination (entero-hepatic recirculation is considered in the model but their parameters are not defined), or distribution. Those parameters are apparently free, which gives little relevance to the results obtained from the model. If this is not the case, and the model parameters are instead well defined and fixed (or varied but explicitly indicated and with reasonable and well supported values), this must be provided in the manuscript or direct and original references must be provided.
2 – Confusing and sometimes inaccurate values assumed for some parameters. For example:
a)    The values considered for the intestinal permeability coefficient in Table 1 (1.56 10^-5 and 9.95 10^-6 for the fasted and fed states respectively) is not in agreement with the values provided in the text (line 248), and none of the parameter sets agrees with the statement “the adapted permeability for fed simulations was 6-fold lower compared to the fasted state”. Additionally, the same permeability is considered for the metabolites in both states (Table 1).
b)    The values considered for the KM and Ki of Ketoconazole and their metabolites for P-gp and CYP3A4 is not in agreement with the parameters found in the reference indicated, nor with the parameters obtained by other authors (Achira M et al, AAPS Pharmsci(1999):1-1).
c)    No reference is provided for P-gp’s kcat or for its concentration (what are the units of the concentrations in Table S1.1?) although there are several estimates for those parameters available. It is indicated that the parameters were assumed or optimized, relative to what objective?
3 – Incorrect citation of the information provided in the SI. There are several incorrections in the citations to the information in the SI. For example, in line 204 (Table S2.3 could not be found), line 231 (Section S1.4 is not about dissolution), line 2.35 (Section S2.4 does not contain plasma concentration-time profiles), and several other. The manuscript must be carefully read and corrected for consistency and accuracy.
4 – Unexpected and/or unexplained results. In some situations, the model predicts a different plasma concentration-time profile in the fasted and fed states (e.g. Figure 3) while in others the same profile is obtained (e.g. Figure S2.17 and S2.18). Also, different profiles are obtained from the model for the same fed state and drug dose (e.g. Figure 3 plots b and d, or plots a and e). This means that some of the parameters that are left free in the model lead to different profiles, and makes the good agreement between the model predictions and (each specific) experimental results rather meaningless.

The above concerns must be clarified before a proper evaluation of the relevance of the work reported in this manuscript may be done.

Round 2

Reviewer 3 Report

please see file attached, text in blue.
